# Cutaneous Neufibroma in the Absence of Classical NF1 Features: A Case Report and Literature Review

**DOI:** 10.3390/dermatopathology12040037

**Published:** 2025-10-15

**Authors:** Christine Suryani Novelita Sutrisno, Desy Hinda Pramita, Ita Puspita Dewi

**Affiliations:** Department of Dermatology and Venereology, Dr. Mohamad Soewandhie General Hospital, Surabaya 60142, Indonesia; desyhinda@gmail.com (D.H.P.); itapuspitadewispkk@gmail.com (I.P.D.)

**Keywords:** neurofibromatosis type 1, cutaneous neurofibroma, atypical presentation, case report

## Abstract

Neurofibromatosis type 1 (NF1) is a prevalent neurocutaneous illness resulting from mutations in the NF1 gene, usually diagnosed according to clinical criteria set by the National Institutes of Health (NIH). These encompass café-au-lait macules, axillary freckling, Lisch nodules, ocular gliomas, osseous lesions, neurofibromas, and familial history. Atypical instances exhibiting partial or isolated characteristics, such as numerous cutaneous neurofibromas (cNFs) absent other classical manifestations, provide a diagnostic difficulty and may be little acknowledged in clinical environments. We describe a 47-year-old male with several soft, non-tender, pinkish-red papules and nodules dispersed throughout the face, torso, limbs, and back. A solitary café-au-lait macule measuring 3 x 2 cm was seen below the right breast, no axillary or inguinal freckling was observed, Lisch nodules were absent during ophthalmologic examination, and there was no pertinent family history. The histopathological examination of a skin lesion verified the diagnosis of cutaneous neurofibroma. According to the NIH guidelines, the patient did not satisfy the requirements for a conclusive diagnosis of NF1. This instance underscores the clinical intricacy of NF1 spectrum diseases and suggests the potential for mosaic NF1 or a minor phenotypic variation. The existence of several cNFs without systemic involvement undermines the adequacy of existing diagnostic paradigms, particularly in adults who exhibit no early-life signs. The psychosocial challenges linked to widespread cNF distribution highlight the necessity for a comprehensive assessment. Limitations encompass the lack of genetic testing, which would have facilitated the confirmation of the diagnosis and the assessment of probable mosaicism. Isolated cutaneous neurofibromas, devoid of other conventional NF1 characteristics, are an uncommon yet clinically pertinent manifestation. Clinicians must uphold a heightened level of suspicion for aberrant NF1 phenotypes and contemplate further examination, using molecular diagnostics where feasible. Reevaluating diagnostic criteria to include these polymorphisms is essential for prompt identification, effective care, and enhanced patient outcomes.

## 1. Introduction

Neurofibromatosis type I (NF1) is a neurocutaneous illness defined by the inactivation of the NF1 (neurofibromin) tumor suppressor gene, resulting from either a de novo mutation or autosomal dominant inheritance [1]. The genetic modification results in a varied array of manifestations that can be clinically identified by at least two or more of the following characteristics: (1) six or more café-au-lait macules, (2) two or more neurofibromas or one plexiform neurofibroma, (3) freckling in the axillary or inguinal regions, (4) Lisch nodules (iris hamartomas), (5) optic gliomas, and (6) osseous lesions [2].

Neurofibromas, including cutaneous (dermal) neurofibromas and plexiform neurofibromas, originate from the biallelic loss of NF1 in Schwann cell lineage [1,3,4]. The cutaneous neurofibroma (cNF) is a neoplasm originating from peripheral nerve Schwann cells, manifesting as a soft nodule inside the dermis of the skin at nearly any anatomical site on the body [5]. The plexiform neurofibroma is present in over 30% of individuals with NF1 and has a risk of transforming into a malignant peripheral nerve sheath tumor, which is associated with a poor 5-year survival prognosis [6]. Conversely, the cNF is observed in over 95% of individuals with the condition, manifesting as soft, skin-colored nodules ranging from 2 mm to 3 cm, with quantities varying from tens to thousands [7]. They are benign histological entities composed of several cell types, devoid of any potential for malignant transformation [8].

Although innocuous, individuals with NF1 see cNF as the most onerous aspect of the condition. Neurological manifestations including irritation, discomfort, and pruritus [7]. Inadequate drying following exposure to moisture can result in issues such as maceration, dermal deterioration, and superficial infections. Physical deformity arises from the presence of hundreds to thousands of cutaneous neurofibromas (cNF) on a person [9]. Evidence associates cNF with worse quality of life stemming from feelings of humiliation, disruption of daily activities such as shopping, challenges in expressing affection towards partners, sexual problems, and negative social consequences. Individuals with NF1 may experience diminished socioeconomic status due to reduced self-esteem and heightened risk aversion, with around fifty percent of those affected suffering from severe depressive illness, possibly exacerbated by their cNF load [10].

Despite the established NIH diagnostic criteria for Neurofibromatosis type I (NF1), the clinical heterogeneity of the condition presents considerable challenges, especially in instances where patients exhibit multiple cutaneous neurofibromas (cNFs) but do not display other classical features such as Lisch nodules, axillary freckling, optic gliomas, or a positive family history. Although few, such occurrences are clinically pertinent and may be disregarded or inaccurately diagnosed as random, non-syndromic skin lesions. Without evident systemic indicators, these individuals may not receive referrals for genetic testing or additional multidisciplinary assessment, so forfeiting the chance for thorough monitoring and early identification of issues. The diagnostic ambiguity intensifies when only one or none of the supplementary NIH criteria are satisfied, compelling clinicians to depend on clinical judgment in circumstances when confirmatory genetic testing may be inaccessible or prohibitively expensive. Moreover, several practitioners may undervalue the significance of solitary cNFs, especially in the absence of pain or neurological impairments, hence postponing a conclusive diagnosis and subsequent treatment.

This case report and literature analysis seek to highlight the clinical and psychological ramifications of isolated cutaneous neurofibromas within the framework of suspected NF1 spectrum illnesses. This study underscores the importance of identifying and examining unusual NF1 presentations, especially in adult patients who may have been inadequately diagnosed in infancy due to mild or missing characteristics. The burden of cNFs, although histologically benign, transcends physical symptoms, frequently resulting in significant mental suffering, social stigmatization, and a reduced quality of life. These lesions can disrupt interpersonal relationships, sexual health, career prospects, and general psychological well-being. Recent consensus revisions to the NF1 diagnostic criteria highlight that mosaic and attenuated forms may not fulfill traditional NIH thresholds, underscoring the need for broader recognition of atypical cases [11]. This case advocates for expanded diagnostic consideration and urges healthcare practitioners to sustain a heightened suspicion for NF1, even when not all diagnostic criteria are met. This method may provide earlier detection, tailored therapy options, and enhanced long-term results for patients who may otherwise be overlooked under the existing diagnostic paradigm.

## 2. Case Description

We report a 47-year-old male presented to the Dermatology and Venereology Clinic with complaints of lumps all over his body. The first lump appeared on his torso about a year ago, and over the past two months, the lumps have spread across his entire body, including the face, arms, and legs. The lumps are neither itchy nor painful. The patient has not experienced any weight loss and reported no visual disturbances. He stated there is no family history of similar conditions. He has no past medical history and does not take any regular medications. He works in an office.

On physical examination as seen in Figure 1, multiple pinkish-red papules were observed, with variable sizes averaging 0.5 to 1 cm in diameter. The nodules were solid and immobile. A single café-au-lait spot measuring 3 × 2 cm was noted below the right breast. No intraoral nodules were found. Due to suspicion of neurofibroma, the patient was referred to an ophthalmology specialist to check for the presence of Lisch nodules. The examination showed no Lisch nodules.

A punch biopsy was performed on one of the papules located on the back. The area was first sterilized using hydrogen peroxide. Local anesthesia was administered using lidocaine with a 1cc intradermal syringe, followed by a punch biopsy, and the specimen was sent for pathological examination. Histopathological evaluation confirmed the diagnosis of cutaneous neurofibroma. The lesion demonstrated spindle-shaped Schwann cells in a collagenous stroma without atypia or malignancy as seen in Figure 2. Immunohistochemical staining (e.g., S100, SOX10, CD34) was not performed due to resource limitations, which is acknowledged as a diagnostic limitation.

## 3. Discussion

The case of a 47-year-old individual with extensive cutaneous neurofibromas (cNFs) without other characteristic characteristics of Neurofibromatosis type 1 (NF1) provides significant clinical insight into the variability and diagnostic complexities of this inherited condition. NF1 is conventionally diagnosed using established NIH criteria, which encompass a mix of café-au-lait macules, axillary or inguinal freckling, Lisch nodules, ocular gliomas, osseous lesions, neurofibromas, and familial history. This patient displayed several cutaneous neurofibromas and a solitary café-au-lait macule, without any other indicators such as Lisch nodules or a hereditary pattern. This incomplete presentation challenges the stringent dependence on criteria that may inadequately include the phenotypic diversity of NF1, particularly in adult-onset or mild instances.

Cutaneous neurofibromas are among the most prevalent and conspicuous symptoms of NF1, usually emerging throughout adolescence and proliferating with age. These are benign tumors derived from Schwann cells, characterized by biallelic inactivation of the NF1 tumor suppressor gene. Despite being histologically benign, they can exhibit considerable variation in both quantity and size, potentially leading to substantial physical and mental consequences. The fast advancement of lesions within a year, especially in cosmetically sensitive regions like the face and neck, certainly created a psychological burden despite the lack of physical discomfort. This case is clinically noteworthy due to the absence of additional traditional NF1 indications, suggesting the potential for an alternate diagnosis within the NF1 spectrum, such as mosaic NF1 or segmental neurofibromatosis.

Mosaic or segmental NF1 develops when a harmful mutation in the NF1 gene emerges post-zygotically, resulting in somatic mosaicism. This indicates that just a portion of cells have the mutation, leading to a localized or partial phenotype. In certain instances, this may present as a localized distribution of neurofibromas in certain body regions or, as may be the case here, a diffuse presentation without systemic involvement. Mosaicism is well-documented in the literature; however, it remains underrecognized, particularly when genetic testing is not conducted. Ruggieri and Huson (2001) delineated many manifestations of mosaic NF1, encompassing pigmentary alterations and neurofibromas in the absence of familial history [2]. Clinical studies demonstrate that patients with mosaic NF1 may still develop systemic complications and therefore benefit from long-term multidisciplinary follow-up similar to generalized NF1 [12].This theory provides a credible foundation for comprehending the restricted yet unique presentation in our case, especially in the absence of hereditary characteristics or systemic involvement.

The biology of cNF is complicated, comprising several cellular components that interact disorganizedly with the extracellular matrix [13,14]. A nerve is essential for the proliferation, growth, and maintenance of NF1-deficient Schwann cells via the perineural milieu that secretes factors like Neuregulin 1 (NRG1) [13]. Immune cells are crucial components of cNF formation. Mast cells are histological indicators of cNF and are attracted to cNF by kit-receptor activation, which facilitates their movement [15]. Mast cell degranulation, induced by trauma or other processes, produces histamine, serotonin, transforming growth factor beta (TGF-B), and other neurotransmitters that may be crucial for the creation and maintenance of cNF [16]. Macrophages, the phagocytic leukocytic immune cells, are found in cNF; however, their role in the progression of disease remains unclear. Fibroblasts are prevalent in the cNF and respond to TGF-B from mast cells by depositing abundant, disordered collagen and undergoing continuous restructuring [17]. Significantly, neurofibroma-associated fibroblasts have distinct characteristics compared to their fibroblast counterparts in keloids or scar tissues since they do not express conventional markers such smooth muscle actin [18]. Additional cell types, including keratinocytes, melanocytes, and adipocytes, are present surrounding cNF but are not deemed essential for their growth [18]. The pathogenesis mechanism remains incompletely elucidated; nonetheless, the predominant explanation posits a maladaptive reaction to molecular or physical stress characterized by an overactive immune response and excessive fibrosis, occurring in the context of NF1 tumor suppression inactivation inside neoplastic Schwann cells. Recent preclinical studies have also developed advanced in vitro and in vivo models of cutaneous neurofibromas, including organoids, genetically engineered mice, and patient-derived xenografts, which provide important platforms for translational research and therapeutic exploration [19].

The anatomical categorization of cNF is organized by stage based on visual characteristics [20,21]. In its first phase, cNF is imperceptible to the naked eye; nevertheless, ultrasonography or other imaging modalities can identify the dermal mass [14]. The cNF is categorized as flat when its skin presentation exhibits hyperpigmentation or little epidermal thinning. The sessile stage of cNF is characterized by the presence of a palpable papule on the skin. It then progresses to the globular stage, characterized by a bigger nodule of 20–30 mm in height and a similar base diameter. The ultimate stage is the pedunculated phase, characterized by the protrusion of dermal cNF contents into a mass elevated above the skin, connected by a stalk.

At now, there is no definitive therapy for cNF. Physical removal is the most efficacious approach for addressing cNF. Physical removal may include techniques such as surgical excision with primary closure and modified biopsy removal methods, or destruction via CO2 laser, electrodessication, and ablation [21,22]. Emerging systemic therapeutic strategies, including MEK inhibitors and targeted molecular therapies, are being explored as promising options to address the tumor burden in NF1 [23]. Obstacles to removal include tumor recurrence due to inadequate excision, substantial scarring, and financial load. The cost remains elevated as cNF is predominantly categorized as an elective, cosmetic procedure by the majority of insurance providers. Moreover, physical removal does not prevent the production of cNF, which can be enhanced with pharmacological treatments.

The NF1 gene on chromosome 17q11.2 encodes neurofibromin, a protein essential for the negative regulation of Ras-mediated cell proliferation and differentiation. The loss or malfunction of neurofibromin results in the constitutive activation of Ras pathway signaling, leading to unregulated cellular proliferation, particularly in neural crest-derived cells like Schwann cells. This unregulated activity may lead to the development of neurofibromas, including both cutaneous and plexiform types. The extensive emergence of cNFs in our patient, despite the absence of systemic involvement, may indicate that the NF1 mutation selectively impacted Schwann cell precursors while preserving other cell types commonly associated with NF1 manifestations, such as melanocytes, which are responsible for café-au-lait spots and Lisch nodules.

The significance of genetic variation in NF1 is paramount. Over 3000 unique variants in the NF1 gene have been found, each possibly linked to a specific phenotype. Certain mutations may produce shortened or nonfunctional neurofibromin, whilst others may disrupt splicing or cause a partial loss of function. This patient may possess a milder or tissue-specific NF1 mutation that remains uncharacterized, especially considering his solitary dermatological manifestation. In the absence of genetic data, it is speculative yet crucial to consider molecular diversity when addressing atypical NF1 patients. Messiaen et al. (2011) highlighted the relationship between mutation type and clinical severity, noting that certain in-frame deletions or splicing mutations lead to attenuated symptoms akin to the one seen here [5].

The psychological ramifications of cutaneous neurofibromas, however frequently being downplayed in the absence of malignancy or neurological issues, demand significant consideration. Multiple studies indicate that persons with a high cNF burden encounter markedly worse quality of life, struggling with self-esteem, interpersonal relationships, sexual activity, and social integration. Mautner et al. (2008) reported the incidence of depressive symptoms and social disengagement in NF1 patients with significant cutaneous involvement, irrespective of internal tumor load [24]. In this instance, despite the patient indicating the absence of pain or pruritus, the visual deformity caused by several visible lesions is likely to impact psychological well-being and social functioning—elements frequently overlooked in the treatment of NF1.

An additional point of interest is the adult-onset or delayed identification of symptoms in this patient. The majority of persons with NF1 are diagnosed during childhood as characteristics such as café-au-lait spots and learning difficulties typically present early. Nonetheless, accumulating research indicates that certain individuals with moderate or atypical phenotypes may remain undetected until maturity, particularly in areas with little access to dermatological or genetic knowledge. Ferner et al. (2007) emphasized the significance of recognizing late-onset NF1 and its ramifications for patient management and monitoring [2]. The postponed onset or diagnosis in this instance may indicate wider systemic problems in healthcare provision and knowledge, particularly in underfunded environments.

This instance also presents significant diagnostic and categorization inquiries. How should physicians categorize a patient with several cutaneous neurofibromas and a solitary café-au-lait macule, without any further diagnostic characteristics? Is it adequate to see them as exhibiting a forme fruste of NF1, or should we establish a new sub-category within the spectrum of NF1-related disorders? Certain research has designated these patients as possessing “non-syndromic multiple neurofibromas” or “atypical NF1,” although a consensus remains absent. Large-scale genotype-first studies have shown that NF1 pathogenic variants are more prevalent than previously recognized, with a significant proportion presenting as mosaic cases that often remain clinically underdiagnosed [25]. Tinschert et al. (2000) observed the presence of similar individuals and advocated for a more comprehensive approach to NF1 diagnosis, one that does not depend exclusively on the NIH criteria, particularly in the lack of molecular evidence [26].

A drawback of this instance is the lack of molecular validation by NF1 gene sequencing. Genetic testing would have offered conclusive information on the presence of an NF1 mutation in this patient and, if present, whether it is of germline or mosaic origin. Regrettably, such testing continues to be costly and unattainable in numerous clinical settings. Furthermore, the absence of long-term follow-up constrains our capacity to observe illness progression or the development of new diagnostic criteria over time. This follow-up might determine if this is a static phenotype or a component of a more extensive, gradually changing NF1 manifestation.

The diagnostic complexities of NF1 in such instances are significant. Misclassification may provide substantial clinical ramifications. Patients may be erroneously persuaded that their illness is benign and localized, so forgoing the chance for systemic monitoring for ocular gliomas, internal neurofibromas, skeletal anomalies, or hypertension—all problems that may develop insidiously in NF1. Furthermore, the genetic ramifications for relatives remain ambiguous without a conclusive diagnosis. In cases with mosaic NF1, germline transfer to progeny is possible if gonadal tissue is involved, highlighting the necessity for reproductive counseling.

This case report has several important limitations. Only one lesion was biopsied, which may not fully represent the disease process, and immunohistochemical stains such as S100, SOX10, and CD34 could not be performed. No molecular genetic testing was available to confirm NF1 mutation or mosaicism, and the absence of such analysis makes the diagnosis of mosaic NF1 speculative. Although MRI facilities are available at the institution, advanced imaging was not performed in this case due to systemic limitations in access to diagnostic procedures. This constraint should be considered when interpreting the case. In addition, no long-term follow-up data are available to assess the clinical progression. These diagnostic and systemic constraints highlight the challenges of managing atypical NF1 presentations in resource-limited settings.

This instance underscores the necessity for a thorough, multidisciplinary strategy for individuals with unexplained neurofibromas. Alongside dermatological assessment, referrals to ophthalmology, neurology, and medical genetics are necessary. Imaging investigations may be warranted, contingent upon the clinical circumstances, to exclude internal neurofibromas or other hidden symptoms. Moreover, doctors must recognize the need for psychological support and patient education, since numerous persons with visible cNFs endure stigma and mental turmoil in silence.

Due to the constraints of existing diagnostic frameworks, there is an immediate necessity for updated recommendations that incorporate clinical, histological, and molecular criteria. A tiered diagnostic strategy, wherein clinical factors inform early suspicion and are augmented by genetic testing when accessible, may assist in reconciling the disparity between underdiagnosis and overdiagnosis. This technique would provide enhanced patient categorization by risk, inform suitable surveillance strategies, and augment long-term results.

This instance underscores the need to record and disseminate unusual manifestations of established illnesses. Each instance contributes to the comprehensive understanding of illness phenotypes and aids in identifying potential deficiencies in existing diagnostic models. In genetic illnesses characterized by significant diversity, such as NF1, case reports are an essential source of clinical knowledge and a foundation for future research hypotheses.

## 4. Conclusions

This case report emphasizes an unusual manifestation of numerous cutaneous neurofibromas without other conventional characteristics of Neurofibromatosis type 1 (NF1), highlighting the phenotypic diversity and diagnostic intricacy of the disorder. The lack of characteristic features such as axillary freckling, Lisch nodules, and familial history suggests the potential for mosaic NF1 or a mild version of the condition, which may be overlooked in clinical practice. Considering the emotional cost and the risk of systemic problems, prompt diagnosis and multidisciplinary assessment are crucial, especially in borderline or incomplete cases. This instance underscores the necessity to re-evaluate existing diagnostic criteria, integrate molecular diagnostics where feasible, and uphold a heightened level of suspicion to guarantee prompt diagnosis, suitable monitoring, and comprehensive therapy for all patients within the NF1 spectrum.

## Figures and Tables

**Figure 1 dermatopathology-12-00037-f001:**
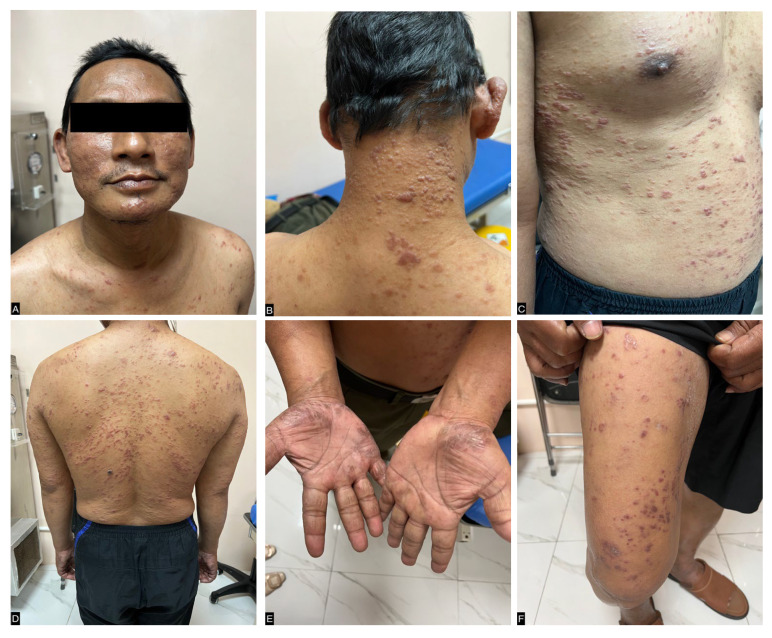
Clinical presentation of a 47-year-old male with multiple cutaneous neurofibromas. (**A**) Frontal view of the face showing multiple small, pinkish to violaceous papules and nodules. (**B**) Posterior view of the neck displaying clusters of dome-shaped papules with no signs of ulceration or inflammation. (**C**) Anterior view of the chest and abdomen revealing numerous scattered papules with a visible café-au-lait macule measuring approximately 3 × 2 cm located below the right breast. (**D**) Posterior view of the back showing widespread distribution of nodular lesions with varying sizes. (**E**) Palmar view of both hands revealing the presence of multiple hyperpigmented macules consistent with palmar freckling. (**F**) Anterolateral view thigh, highlighting similar lesions extending to the lower extremities.

**Figure 2 dermatopathology-12-00037-f002:**
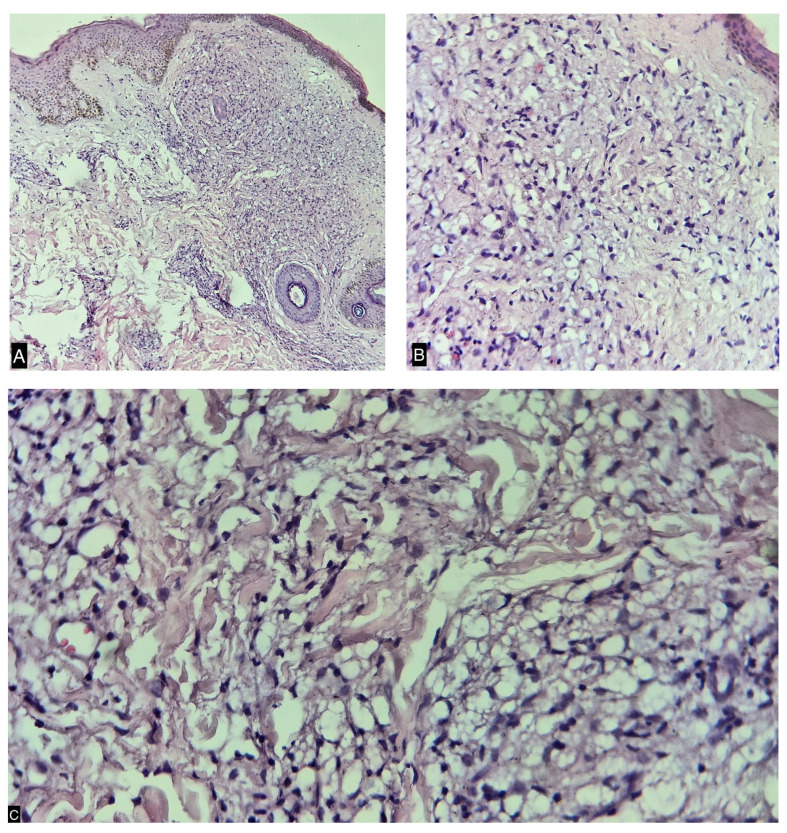
Histopathological examination of a cutaneous neurofibroma lesion using Hematoxylin–Eosin (H&E) staining Result shows a section of tissue covered with squamous epithelium. In the stroma, clusters of cells with oval to spindle-shaped nuclei are seen, some arranged in a wavy pattern, with fine chromatin, arranged in alternating sheets with fibrous connective tissue. No signs of malignancy observed. (**A**) 100× magnification (**B**) 200× magnification (**C**) 400× magnification.

## Data Availability

The data are not publicly available due to patient confidentiality. Data related to this study has been included in this manuscript.

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
