# Peer review of "Cutaneous Neufibroma in the Absence of Classical NF1 Features: A Case Report and Literature Review"

_dermatopathology, 2025, doi:10.3390/dermatopathology12040037_

Round 1
Reviewer 1 Report
Comments and Suggestions for Authors
I read with great interest the manuscript. However, I have some comments:
-did you do any imaging to your patient? Eg Xray/MRI etc for osseous lesions or optic gliomas?
-have you done any more stains other than H&E? eg S100, SOX10, CD34
-In line 122 please add the diagnosis and a small description before the figure 2
-You should add more recent publications
-Your case is really interesting, particularly given the late and rapid onset of neurofibromas. Were you able to identify any possible triggering factor? A limitation as you also mentioned is the absence of molecular confirmation via NF1 gene sequencing, which could have strengthened the diagnostic insight.
Overall, this article is a solid effort. With some revisions, it can be greatly enhanced.
Author Response
We sincerely thank Reviewer 1 for the thoughtful and constructive comments, which have greatly helped us improve the clarity and quality of our manuscript.
Comment 1: Did you do any imaging (X-ray/MRI) for osseous lesions or optic gliomas?
Response: We acknowledge this limitation. As clarified in the Discussion (line 285), although MRI facilities are available at the institution, advanced imaging was not performed due to systemic limitations in access to diagnostic procedures. This constraint has been explicitly stated in the revised manuscript. Also due to Indonesian national health insurance program diagnostic procedure are limited
Comment 2: Have you done any stains other than H&E (e.g., S100, SOX10, CD34)?
Response: We have added this information in the Case Description (line 126–128), explicitly stating that immunohistochemical staining was not performed due to resource limitations and is acknowledged as a diagnostic limitation.
Comment 3: In line 122 please add the diagnosis and a small description before Figure 2.
Response: Revised as requested. At line 124, we now state: “Histopathological evaluation confirmed the diagnosis of cutaneous neurofibroma. The lesion demonstrated spindle-shaped Schwann cells in a collagenous stroma without atypia or malignancy.”
Comment 4: You should add more recent publications.
Response: We have added updated references between 2021–2025, including Legius et al. (2021) [11], Ejerskov et al. (2021) [12], Staedtke et al. (2023) [19], Park (2024) [23], and Safonov et al. (2025) [24]. Corresponding sentences have been inserted in the Introduction and Discussion (lines 90, 164, 189, 205, 259).
Comment 5: Were you able to identify any possible triggering factor?
Response: We have clarified in the Case Description (line 102–104) that the patient denied any trauma, medication, or infection that could have acted as a triggering factor.
Comment 6: The absence of molecular confirmation via NF1 gene sequencing is a limitation.
Response: This has been clearly restated in the Discussion → Limitations (lines 283–285), stating “No molecular genetic testing was available to confirm NF1 mutation or mosaicism, and the absence of such analysis makes the diagnosis of mosaic NF1 speculative.”
Reviewer 2 Report
Comments and Suggestions for Authors
The authors present a patient who developed multiple cutaneous lesions, mostly over the past two months. The lesions are clinically nonspecific. Only one lesion was studied histologically, and although it could correspond to a neurofibroma, other possibilities such as a variant of histiocytosis cannot be totally excluded without a complete immunophenotypic evaluation. The manuscript also includes a review of the complex diagnosis of NF1, but in some sections it becomes cumbersome, repetitive, and overly philosophical.
The main shortcomings of this case are:
- The late onset of the lesions is an unusual clinical presentation for neurofibromatosis and would require an irrefutable histological diagnosis of the skin lesions.
- Only one of the lesions was biopsied, which may not be representative of the process. At least a second biopsy from a different lesion would be necessary for diagnostic confirmation..
- In a case that is clinically atypical, the histological diagnosis of a neurofibroma should be supported by a complete immunophenotypic evaluation, ruling out other possible entities.
- The text is unnecessarily long. The essential information could be summarized in a few paragraphs.
Comments on the Quality of English Language
The text is well-written but contains some stylistic excesses. The text would benefit from a concise, scientific style.
Author Response
We sincerely thank Reviewer 2 for the insightful and constructive feedback, which has guided us in refining and strengthening the manuscript.
Comment 1: The late onset is unusual and would require irrefutable histology.
Response: We addressed this by emphasizing the rarity of adult-onset NF1 and the limitation of immunohistochemistry in the Discussion (line 280-282). Also due to Indonesian national health insurance program diagnostic procedure are limited.
Comment 2: Only one lesion was biopsied; a second would be preferable.
Response: This limitation is acknowledged in the Discussion (line 280–282), This case report has several important limitations. Only one lesion was biopsied, which may not fully represent the disease process, and immunohistochemical stains such as S100, SOX10, and CD34 could not be performed. Also due to Indonesian national health insurance program diagnostic procedure are limited.
Comment 3: Histological diagnosis should be supported by immunophenotypic evaluation.
Response: As above, we added in the Case Description and Discussion (lines 124–127; 280–282). Again due to Indonesian national health insurance program diagnostic procedure are limited.
Comment 4: The text is unnecessarily long and repetitive.
Response: In the Discussion section, we also want to provide a literature overview of NF1 and relate it to the present case.
Reviewer 3 Report
Comments and Suggestions for Authors
Congratulations to the authors of the article!
The authors report a 47 year old patient with numerous cutaneous neurofibromas but without hallmark NF features. This case highlights the diagnostic complexity of NF1 and the limitations of current criteria, particularly in cases with adult-onset or mild phenotypes. The report is strengthened by its comprehensive integration of literature and well-documented clinical and histological findings. The discussion section is exhaustive and very well put.
There are certain weaknesses that I would like to mention. The absence of genetic testing means that the diagnosis of mosaic NF1 remains speculative, and no long-term follow-up is included to observe disease progression.
Despite these limitations, the article makes a meaningful contribution to the field by challenging diagnostic frameworks and advocating for broader recognition of NF1 spectrum disorders. It would be improved by incorporating genetic analysis (if possible), sharing follow-up data, and clearer differential diagnosis.
Overall, it is a valuable case report that encourages a multidisciplinary approach to NF1 diagnosis and care, and I favor its publication.
Author Response
We sincerely thank Reviewer 3 for the valuable and detailed comments, which have helped us improve the clarity, depth, and overall quality of the manuscript.
Comment 1: The absence of genetic testing means mosaic NF1 remains speculative.
Response: We agree and emphasized this in the Discussion → Limitations (lines 282–284), No molecular genetic testing was available to confirm NF1 mutation or mosaicism, and the absence of such analysis makes the diagnosis of mosaic NF1 speculative.
Comment 2: No long-term follow-up included.
Response: We have clarified in the Limitations (line 287–289), In addition, no long-term follow-up data are available to assess the clinical progression. These diagnostic and systemic constraints highlight the challenges of managing atypical NF1 presentations in resource-limited settings. Due to Indonesian national health insurance program diagnostic procedure and follow up session are limited.
Comment 3: Clearer differential diagnosis would help.
Response: In Discussion (lines 153), we explain how mosaic NF1 and segmental neurofibromatosis were considered, with reference to Ruggieri and Huson (2001). Histopathology findings are also detailed to distinguish from mimics such as histiocytosis.
Round 2
Reviewer 2 Report
Comments and Suggestions for Authors
The manuscript has been improved by explicitly addressing its limitations. Since the histological features in the image are consistent with neurofibroma, the absence of immunohistochemical stains does not appear to be a major concern. The main issue, however, is relying on a single biopsy to establish the diagnosis of such a serious condition. It would be helpful to emphasize that the patient was either lost to follow-up or declined a repeat biopsy, to highlight the importance of confirming the diagnosis adequately. The text should avoid giving the impression that neurofibromatosis can be reliably diagnosed based on a single biopsy.
Author Response
We sincerely thank Reviewer 2 for the insightful and constructive feedback. Please find our detailed responses below. All corresponding revisions are highlighted in the revised manuscript.
Comment 1: Diagnosis is based on only one biopsy. Please clarify this point, as NF1 cannot be confirmed with a single biopsy.
Response 1: Thank you for pointing this out. We agree with this important concern. In the revised manuscript, we clarified in the Case Description (page 5, lines 124–125) that the patient declined additional procedures. In addition, in the Discussion – Limitations (page 7, lines 282–284), we now explicitly state that only one lesion was biopsied, the patient declined further biopsies, and was subsequently lost to follow-up. This addition acknowledges the diagnostic limitation and emphasizes the need for caution in interpreting the case.
“Only one lesion was biopsied, as the patient declined additional procedures and was subsequently lost to follow-up.” (Case Description, lines 124–125)
“Only one lesion was biopsied, which may not fully represent the disease process, as the patient declined additional biopsies and was subsequently lost to follow-up.” (Discussion – Limitations, lines 282–284)